# Evaluation of a mobile application tool (BiliNorm) to improve care for newborns with hyperbilirubinemia in Indonesia

**Mahendra T. A. Sampurna**[1,2]*, **Kinanti A. Ratnasari**[2], **Zahra S. Irawan**[2], **Risa Etika**[2], **Martono T. Utomo**[2], **Brigitta I. R. V. Corebima**[3], **Pieter J. J. Sauer**[4], **Arend F. Bos**[4], **Christian V. Hulzebos**[4], **Peter H. Dijk**[4]

**1** Department of Pediatrics, Faculty of Medicine Universitas Airlangga, Airlangga Teaching Hospital, Surabaya, Indonesia, **2** Department of Pediatrics, Faculty of Medicine Universitas Airlangga, Dr Soetomo General Hospital, Surabaya, Indonesia, **3** Department of Pediatrics, Faculty of Medicine, Universitas Brawijaya, Dr. Saiful Anwar General Hospital, Malang, Indonesia, **4** Department of Pediatrics, Beatrix Children's Hospital, University Medical Center Groningen, Groningen, The Netherlands

* mahendra.tri@fk.unair.ac.id

**Data Availability Statement:** Data cannot be shared publicly because it contains potentially identifying information and sensitive information.

## Abstract

### Background

Severe hyperbilirubinemia is more frequent in low- and middle-income countries such as Indonesia than in high-income countries. One of the contributing factors might be the lack of adherence to existing guidelines on the diagnosis and treatment of hyperbilirubinemia. We developed a new national guideline for hyperbilirubinemia management in Indonesia. To help healthcare workers use this guideline, a web-based decision support tool application may improve both the adherence to the guideline and the care for infants with hyperbilirubinemia.

### Methods

We developed a web-based application (BiliNorm) to be used on a smartphone that displays the bilirubin level of the patient on the nomogram and advises about the treatment that should be started. Healthcare workers of two teaching hospitals in East Java, Indonesia, were trained on the use of BiliNorm. At 6 months after the introduction, a questionnaire was sent to those who worked with the application enquiring about their experiences. An observational study was conducted in two time epochs. A chart review of infants with hyperbilirubinemia in the two hospitals was sent. The appropriateness of hyperbilirubinemia management during a 6-month period before BiliNorm introduction was compared to that during a 7-month period after its introduction.

### Results

A total of 43 participants filled in the questionnaire, the majority (72%) of them indicated that BiliNorm was well received and easy to use. Moreover, 84% indicated that BiliNorm was helpful for the decision to start phototherapy. Chart review of 255 infants before BiliNorm

Data is available from the Ethical Committee of Dr Soetomo General Hospital. Contact via lit.rsds1@gmail.com for access to confidential data.

**Funding:** This project was supported by a research grant from the National Institute of Health Research and Development (NIHRD), Ministry of Health, Republic of Indonesia, Jakarta, Indonesia HK.03.01/I/1186/2019. The funders had no role in study design, data collection and analysis, decision to publish, or preparation of the manuscript.

**Competing interests:** The authors declare that they have no conflict of interests to declare.

introduction and that of 181 infants after its introduction indicated that significantly more infants had received treatment according to the guideline (38% vs 51%, p = 0.006). Few infants received phototherapy, but bilirubin level was not measured (14% vs 7%, p = 0.024). There was no difference in the proportion of infants who were over- and under-treated (34% vs 32% and 14% vs 10%, respectively).

## Conclusions

The web-based decision tool BiliNorm appears to be a valuable application. It is easy to use for healthcare workers and helps them adhere to the guideline. It improves the care for infants with hyperbilirubinemia and may help reduce the incidence of severe hyperbilirubinemia in Indonesia.

## Background

Severe hyperbilirubinemia is frequently observed in newborn infants in developing countries such as Indonesia [1]. Without adequate treatment, this condition can result in bilirubin-induced acute and chronic encephalopathy and may even cause neonatal death [2]. Although there are guidelines on the diagnosis and treatment of hyperbilirubinemia to assist healthcare workers involved in the care of newborn infants, such as midwives, general practitioners and paediatricians, a previous research showed that the awareness and adherence to these guidelines is low among midwives and general practitioners in Indonesia [3]. In fact, only 23% of general practitioners and 29% of midwives use these guidelines. In a survey conducted at Dr. Soetomo General District Hospital, Surabaya, Indonesia, it was found that 43% of term babies and 59% of near-term babies received phototherapy (PT), but the total serum bilirubin (TSB) level was below the PT threshold. At the same time, under-treatment was found in 30% of pre-term babies. Another study conducted in the USA reported that approximately 60% of paediatricians initiated PT at TSB levels lower than those recommended by the American Academy of Pediatrics (AAP) in newborns aged >72 hours, indicating that paediatricians do not always adhere to guidelines [4]. Over-treatment and under-treatment are problems in hyperbilirubinemia management that can be prevented along with the adherence of healthcare workers to hyperbilirubinemia management guidelines. Low adherence to guidelines by healthcare workers may be due to the difficulty in accessing the available guidelines [3, 5]. Among paediatricians in Surabaya, 50% indicated difficulties in obtaining access to a guideline [3]. Therefore, in this study, we developed the first web-based decision application based on the Indonesian hyperbilirubinemia guideline to make it easier for paediatricians, midwives and general practitioners to access and adhere to the guideline.

This paper is intended to describe the characteristics of this web-based application, known as BiliNorm [6], and how this application was perceived by paediatricians, midwives and general practitioners. We hypothesised that the introduction of BiliNorm influenced the accuracy of treatment of infants with hyperbilirubinemia and would improve the care for these infants.

## Methods

BiliNorm is based on the Indonesian National Guideline on Hyperbilirubinemia [7]. It can be accessed at www.bilinorm.babyhealthsby.org via mobile phone or personal computer with two language options, Indonesian and English [6]. A digital research institute in 2018 estimated

approximately 100.000.000 out of total 250.000.000 population in Indonesia uses smartphone [8], where this penetration rate might be higher among healthcare workers population considering them as a more well-educated group. The BiliNorm application algorithm is depicted in Fig 1.

When opening the BiliNorm application, users have to fill in the following patient data: gestational age (weeks), date and time of birth, date and time of record, birth weight (g), TSB level (mg/dL or μmol/L) as well as risk factors (Fig 2A). The risk factors incorporated in this tool are adapted from the AAP guideline [9] and include ABO/Rhesus incompatibility, haemolysis (G6PD deficiency or spherocytosis), other illness (asphyxia, infection) and hypoalbuminemia (<30 mg/L). When 'no risk factors' is filled in, the case is considered as uncomplicated hyperbilirubinemia. When 'unknown risk factors' is filled in, the patient is categorised as having risk factors (other than those listed or not being able to check the risk factors). After filling in these data, the actual TSB level of the infant is shown in the TSB nomogram that displays the

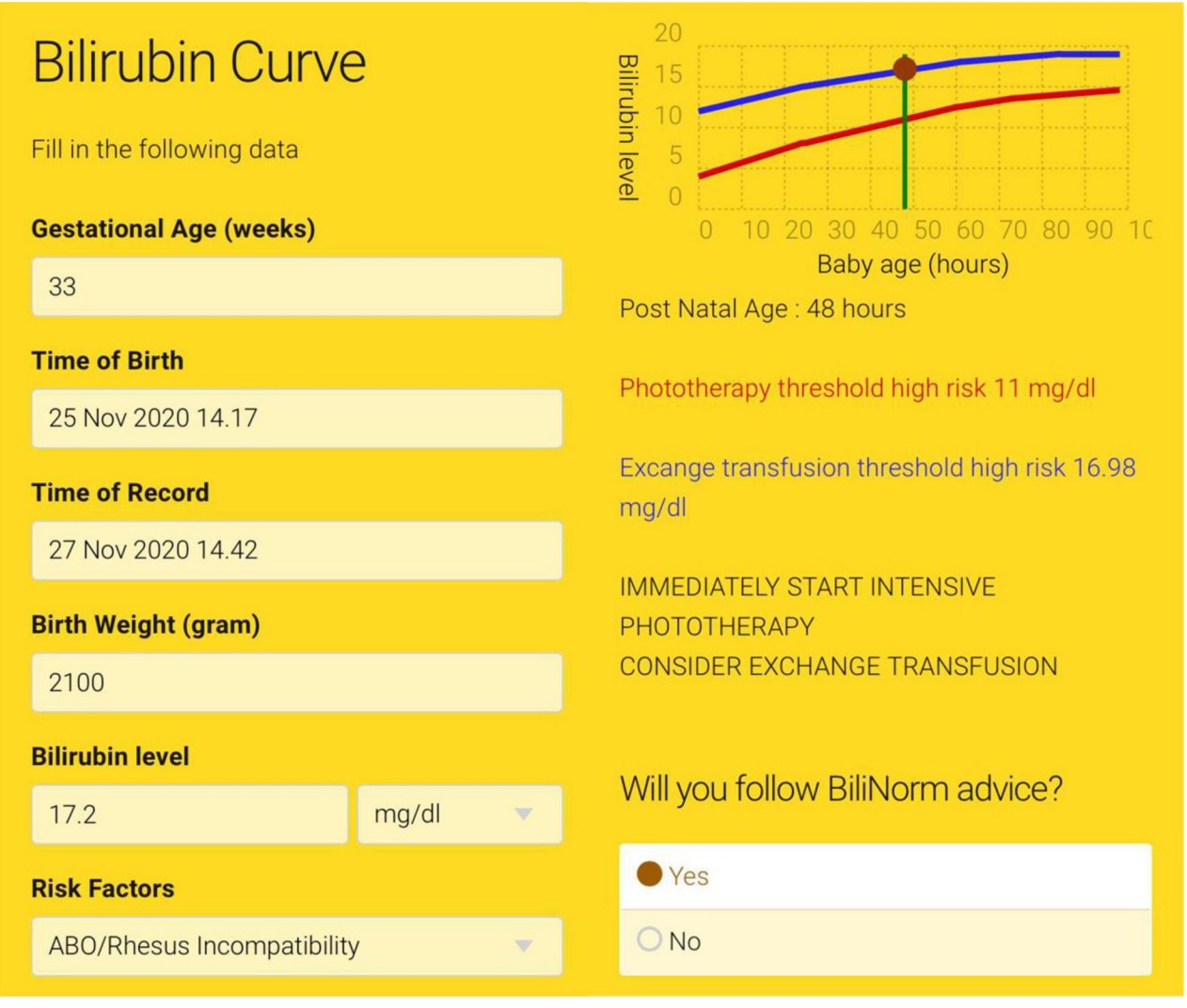

**Fig 1. BiliNorm application algorithm.** (ABE = Acute Bilirubin Encephalopathy, BIND-M = Bilirubin-Induced Neurological Dysfunction-Modified, KSD = Kernicterus Spectrum Disorder, NICE = National Institute for Health and Care Excellence).

treatment thresholds of PT and the exchange transfusion over time. Advice for the caretaker on how to treat the infant is provided as follows: no treatment, start PT or immediately start intensive PT and consider exchange transfusion.

Fig 2 shows an example of data entry in the BiliNorm application of a preterm infant born at 33 weeks of gestation. The infant was admitted to a neonatal unit with jaundice at day 2. Her birth weight was 2100 g. Blood test revealed ABO incompatibility and a TSB level of 17.2 mg/dL. After entering all data, the results showed both the treatment threshold lines of PT and the exchange transfusion for this patient. The recommendation for the paediatrician regarding this patient management is to start immediately with intensive PT and consider exchange transfusion.

Different nomograms are included in BiliNorm, one for infants with a gestational age >35 weeks and four for preterm infants. Infants born after <35 weeks of gestational age have different thresholds compared with term babies according to the new Indonesian National Guideline on Hyperbilirubinemia. In Indonesia, it is often difficult to determine the exact gestational age; therefore, guidelines for preterm babies are categorised into the following birth weight categories: ≤1000, 1001–1499, 1500–1999, and >1999 g.

In addition to the advice regarding the potential treatment for hyperbilirubinemia, information is provided about the risks of complications due to acute bilirubin encephalopathy (ABE). This is based on the modified bilirubin-induced neurological dysfunction-modified (BIND-M) scoring adapted from Radmacher *et al*. [10], which requires the examination of mental status, muscle tone, altered cry and altered gaze. The result is provided in the following four categories: 0, no ABE; 1–4, mild ABE; 5–6, moderate ABE; and >6, severe ABE. In this

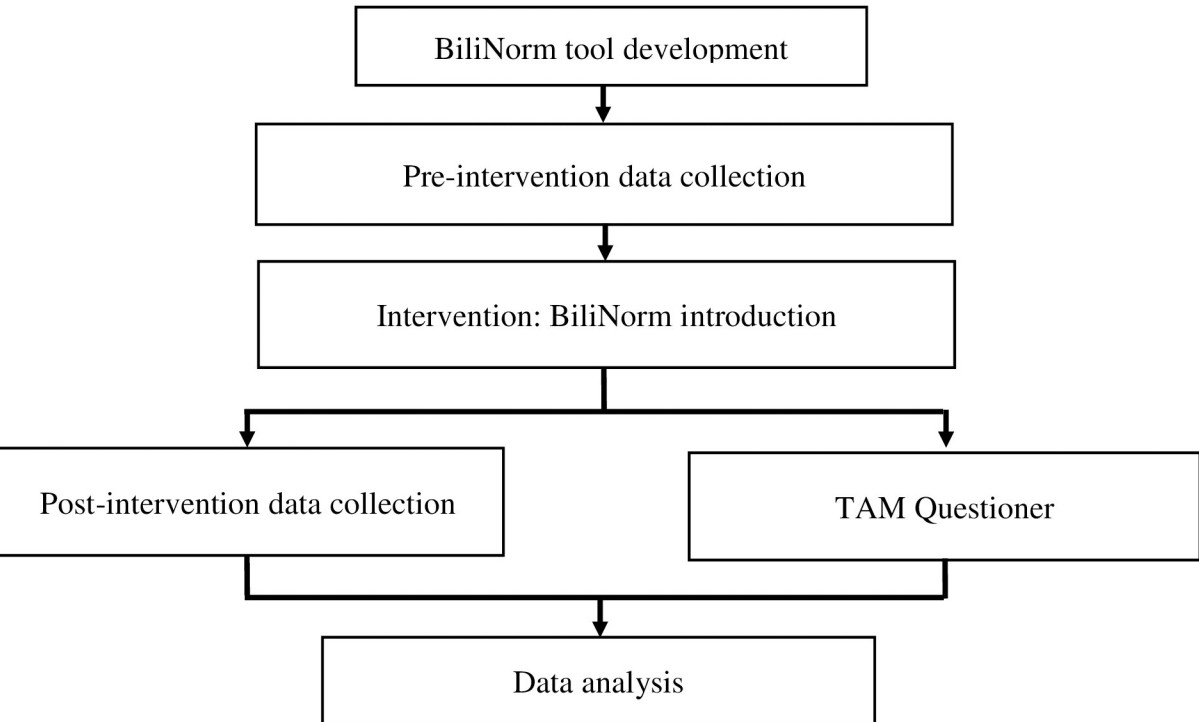

**Fig 2. BiliNorm screenshots.** The screenshots show the data fields required to be filled in on the input screen (A) and the results and advice given in the output screen (B) of BiliNorm. With the permission of Sampurna & Kurniwan, 2019 (https://bilinorm.babyhealthsby.org/) [6].

case, the patient had mild hypotonia and a high-pitched cry, and thus, the BIND-M score of 2 was classified as mild ABE with a probable low risk of neurological complications (if appropriate treatment is provided timely).

Another feature of BiliNorm is the advice for follow-up examination in the outpatient clinic. This is based on the possible diagnosis of kernicterus spectrum disorder (KSD) and consists of a scoring system that includes the highest TSB level, the presence of risk factors, the findings of the neurological examination at first presentation and at follow-up, the presence of enamel dysplasia, the results of the auditory brainstem response (ABR) test and the magnetic resonance imaging findings [11]. The result is provided as one of the following four categories: definite kernicterus (10–14), probable kernicterus (6–9), possible kernicterus (3–5) and no kernicterus (0–2). This feature might help prepare for possible long-term complications of severe hyperbilirubinemia.

Communication to the patient's family is an often neglected and difficult issue for healthcare workers in low- and middle-income countries, especially in Indonesia. Therefore, BiliNorm also provides an educational checklist on what should be told to the patient's family. The checklist is adopted and adapted from the National Institute for Health and Care Excellence (NICE) guideline on neonatal jaundice [12].

This investigation was an observational study conducted in two time epochs in two general district hospitals in East Java, i.e. Dr. Soetomo General Hospital, Surabaya, Indonesia, and Saiful Anwar General Hospital, Malang, Indonesia. Fig 3 shows the research algorithm.

The BiliNorm application was introduced to healthcare workers comprising midwives, paediatric residents and paediatricians in March 2019. The participants were asked to use the application after its introduction.

To evaluate how the BiliNorm application was perceived in practice, questionnaires were sent to healthcare workers who used the application through Google-form. The questionnaire was distributed through internal communication portal which consists of 150 residents and 53 NICU nurses. The questionnaire used in this study was adapted and developed from Davis' Technology Acceptance Model (TAM) [13]; it had the following four major components: 1. perceived usefulness, 2. perceived ease of use, 3. subjective norm and 4. intention to use BiliNorm in the future. There were a total of 22 questions, with each question consisting of seven responses, resulting in a score ranging from 1 (strongly disagree) to 7 (strongly agree). The questionnaire used in this study is shown as S1 Data.

To examine whether the introduction of BiliNorm had an effect on the treatment of infants with hyperbilirubinemia, the medical records of all patients with neonatal hyperbilirubinemia admitted to the neonatal units were collected in both hospitals during a 6-month period before the introduction of BiliNorm (September 2018 to March 2019) and a 7-month period after its introduction (April to September 2019).

Data on gestational age, birth weight, birth date, risk factors, and TSB level were collected to determine the treatment that should have been provided to the patients based on the Indonesian Hyperbilirubinemia Guideline. Next, the actual treatment that was provided was compared with the treatment that was indicated by the guideline. All cases were divided into the following four groups: under-treatment, correct treatment, over-treatment and inappropriate treatment. Under-treatment implied that the infant did not receive any treatment despite the TSB level being above the PT threshold. Over-treatment implied that the infant received PT despite the TSB level being below the PT threshold. Correct treatment was defined as treatment according to the Indonesian Hyperbilirubinemia Guideline. Inappropriate treatment indicated that treatment was provided without any TSB measurement.

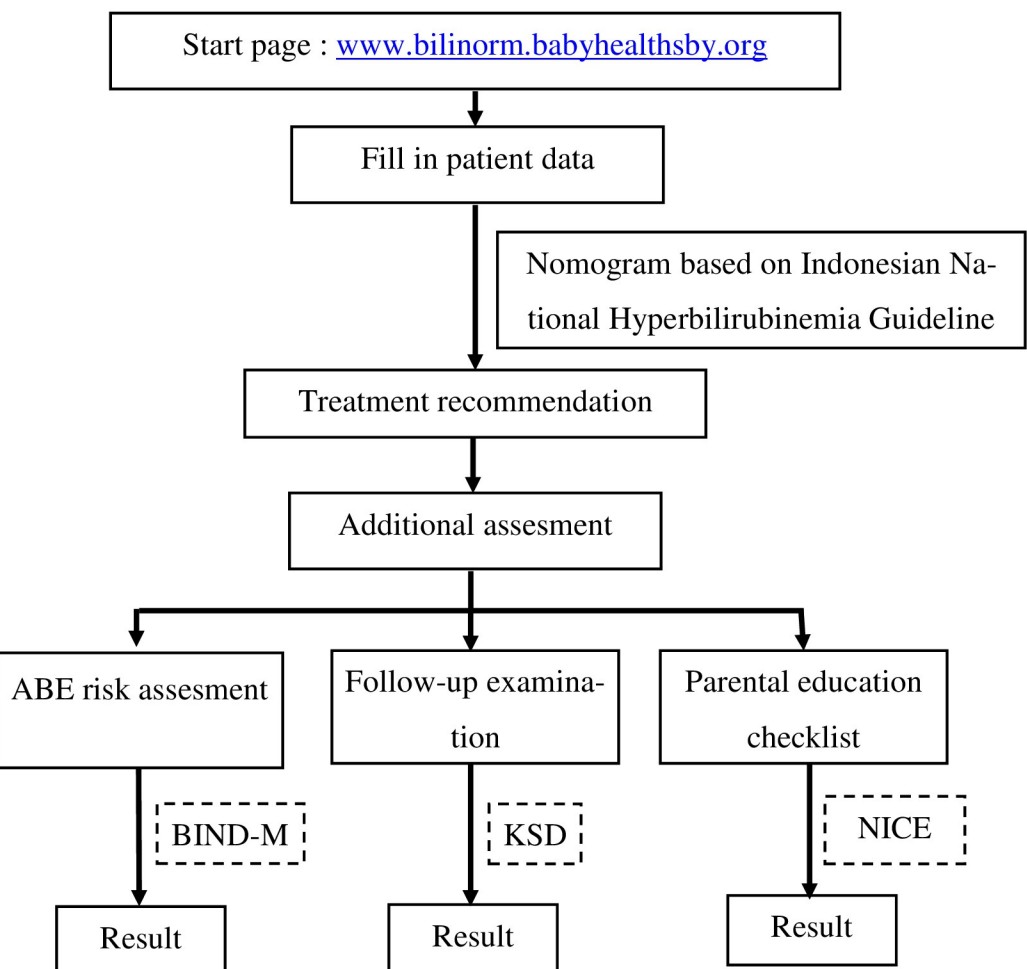

**Fig 3. Research algorithm.** Fig 3 shows the proportion of infants <35 weeks of gestational age and 35 weeks of gestational age categorised according to treatment classification. According to the results, BiliNorm showed a good contribution, especially for infants <35 weeks of gestational age. This can be observed by the increased rate of correct treatment and a reduction of inappropriate treatment in BiliNorm. Abbreviation (ABE = Acute Bilirubin Encephalopathy, BIND-M = Bilirubin Induced Neurological Dysfunction-Modified, KSD = Kernicterus Spectrum Disorder, NICE = National Institute for Health and Care Excellence).

Ethical approval was provided by the Ethical Committee in Health Research of Dr. Soetomo General Hospital, Surabaya (Number 1060/KEPK/III/2019).

## Data and statistical analysis

Data collected from the medical records were analysed using SPSS for Windows, Version 21 (IBM, Corp. Armonk, N.Y., USA). Pearson's chi-square test was used to calculate the p value of the proportion of infants within a gestational age category, birth weight category or having risk factors and treatment classifications for the pre-introduction period compared with the post-introduction period of BiliNorm. The p value of correct treatment, under-treatment, over-treatment and inappropriate treatment was calculated using Pearson's chi-square test for the comparisons of pre-introduction versus post-introduction of BiliNorm. Probability values <0.05 were considered to be statistically significant.

# Results

## Questionnaire study

Table 1 shows all the questions and responses to the questionnaire. In total, 43 users of Bili-Norm returned the questionnaires, however the periodical changes of the number of health-care workers involved in the communication portal made it hard to get the questionnaire's response rate. As much as 56% of the participants strongly agreed and another 28% agreed that the BiliNorm application helped in determining whether it is indicated to start PT in new-born infants with jaundice. The application also provided better quality of hyperbilirubinemia management (56% strongly agreed and 28% agreed). Furthermore, 81% of the respondents agreed or strongly agreed that it helped in providing better communication, information and education to the parents and family regarding hyperbilirubinemia and its effects. In the section 'perceived ease of use, 49% of them strongly agreed and another 23% agreed that it was easy to use BiliNorm. None of the respondents indicated that it was difficult to use. A small minority of 5% of the respondents reported slight disagreement on the three statements on clarity, flexi-bility and skillfulness required to use the application. For the questions regarding the subjec-tive norm, 42% of them strongly agreed and another 28% agreed that their colleagues supposed that BiliNorm is important for the responder. Moreover, 42% of the respondents strongly agreed and 32% agreed to the intention to use BiliNorm in the near future.

## Chart review

We collected 436 medical records from 255 infants with hyperbilirubinemia during the 6-month period before the introduction of BiliNorm and from 181 infants with hyperbilirubi-nemia during the 7-month period after its introduction. Fig 4 shows the demographic data of these infants. The proportion of infants receiving correct treatment was significantly higher after the introduction of BiliNorm than before its introduction (38% vs 51%; p < 0.006). There were few cases of under- and over-treatment, but this number was not statistically significant. The rate of treatment with PT without measuring the TSB level decreased from 14% to 7% (p = 0.024).

Table 2 describes whether BiliNorm contributes to good results as implemented in the teaching hospital. BiliNorm had a good impact in providing suggestions to healthcare person-nel as reported in the pre-post analysis, p = 0.016. The results indicated statistically significant differences between the correct treatment and inappropriate treatment rates before and after BiliNorm implementation.

Fig 4 shows the results for infants with a gestational age above and below 35 weeks. There was no change in the incidence of correct treatment for infants >35 weeks of gestational age, ranging from 44% to 50% (p = 0.393), but the incidence increased for preterm infants <35 weeks of gestational age, ranging from 33% to 53% (p < 0.003). Over- and under-treatment rates did not change significantly in both age groups. The rate of treatment with PT without measuring the TSB level did not change significantly, the proportion being 9%–7% (p = 0.660) in infants >35 weeks of gestational age and decreasing from 18% to 7% (p < 0.014) in infants <35 weeks of gestational age. Out of 93 children who received correct treatment, only 1 child who received exchange transfusion.

# Discussion

In this study, we developed a web-based application, BiliNorm, to be used on a smartphone to improve the usage of the newly developed Indonesian guideline on the diagnosis and treat-ment of hyperbilirubinemia. A survey conducted on a group of healthcare workers who were

**Table 1. Technology acceptance model results of the BiliNorm questionnaire.**

| Items | | Answer N (%) | | | | | | |
|---|---|---|---|---|---|---|---|---|
| | | 1 | 2 | 3 | 4 | 5 | 6 | 7 |
| **A. Perceived Usefulness** | | | | | | | | |
| 1. | BiliNorm helps you quickly to decide the need of phototherapy in jaundiced babies | 0 | 0 | 1 (2) | 5 (12) | 1 (2) | 12 (28) | 24 (56) |
| 2. | BiliNorm helps you to be more aware of Acute Bilirubin Encephalopathy | 0 | 0 | 0 | 6 (14) | 2 (5) | 11 (25) | 24 (56) |
| 3. | BiliNorm helps you to be more aware of kernicterus | 0 | 0 | 0 | 4 (9) | 2 (5) | 12 (28) | 25 (58) |
| 4. | BiliNorm helps you improve your hyperbilirubinemia management | 0 | 0 | 0 | 4 (9) | 3 (7) | 12 (28) | 24 (56) |
| 5. | BiliNorm helps you to improve communication, information, and education to parents about hyperbilirubinemia and its effects | 0 | 0 | 0 | 6 (14) | 2 (5) | 13 (30) | 22 (51) |
| 6. | BiliNorm helps you to improve the follow-up for hyperbilirubinemia babies | 0 | 0 | 0 | 4 (9) | 2 (5) | 14 (33) | 23 (53) |
| **B. Perceived ease of use** | | | | | | | | |
| 1. | Learning to use BiliNorm is easy for you | 0 | 0 | 0 | 6 (14) | 6 (14) | 10 (23) | 21 (49) |
| 2. | You find BiliNorm is easy to get the Information that you want to improve you hyperbilirubinemia management | 0 | 0 | 0 | 8 (19) | 4 (9) | 10 (23) | 21 (49) |
| 3. | You find BiliNorm is clear and understandable | 0 | 0 | 2 (5) | 4 (9) | 2 (5) | 15 (35) | 20 (46) |
| 4. | You find BiliNorm is flexible to use | 0 | 0 | 2 (5) | 3 (7) | 7 (16) | 10 (23) | 21 (49) |
| 5. | It is easy for you to become skillful in using BiliNorm | 0 | 0 | 2 (5) | 3 (7) | 6 (14) | 13 (30) | 19 (44) |
| 6. | You find BiliNorm easy to use | 0 | 0 | 0 | 5 (12) | 5 (12) | 11 (25) | 22 (51) |
| **C. Subjective Norm** | | | | | | | | |
| 1. | Your colleagues think that BiliNorm is important to you | 0 | 0 | 2 (5) | 6 (14) | 5 (11) | 12 (28) | 18 (42) |
| 2. | It is important to your colleagues that you continue to use BiliNorm | 0 | 0 | 1 (2) | 8 (19) | 4 (9) | 13 (30) | 17 (40) |
| 3. | It would not really matter to your colleagues if you stopped using BiliNorm | 2 (4) | 2 (4) | 3 (7) | 9 (21) | 8 (19) | 7 (16) | 12 (28) |
| 4. | Your colleagues would expect you to continue to use BiliNorm | 0 | 1 (2.3) | 1 (2.3) | 6 (14) | 6 (14) | 13 (30) | 16 (37) |
| 5. | None of your colleagues would really be surprised if you stopped using BiliNorm | 0 | 1 (2) | 3 (7) | 12 (28) | 5 (12) | 9 (21) | 13 (30) |
| 6. | Your colleagues would probably be disappointed in you if you stopped using BiliNorm | 0 | 2 (5) | 0 | 13 (30) | 4 (9) | 11 (26) | 13 (30) |
| 7. | Your colleagues would probably make you feel guilty if you stopped using BiliNorm | 0 | 3 (7) | 1 (2) | 11 (26) | 7 (16) | 10 (23) | 11 (26) |
| **D. Intention to use BiliNorm** | | | | | | | | |
| 1. | You intend to use BiliNorm in the next months | 0 | 0 | 0 | 5 (12) | 6 (14) | 14 (32) | 18 (42) |
| 2. | You predict that you would use BiliNorm in next months | 0 | 0 | 1 (2) | 6 (14) | 5 (12) | 12 (28) | 19 (44) |
| 3. | You plan to use BiliNorm in the next months | 0 | 0 | 1 (2) | 6 (14) | 4 (9) | 14 (33) | 18 (42) |

The questions are translated from Bahasa Indonesia into English. Data are presented as numbers and (percentages). Answers categories: 1. Strongly disagree, 2. Disagree, 3. Slightly disagree, 4. Neither agree nor disagree, 5. Slightly agree, 6. Agree, 7. Strongly agree.

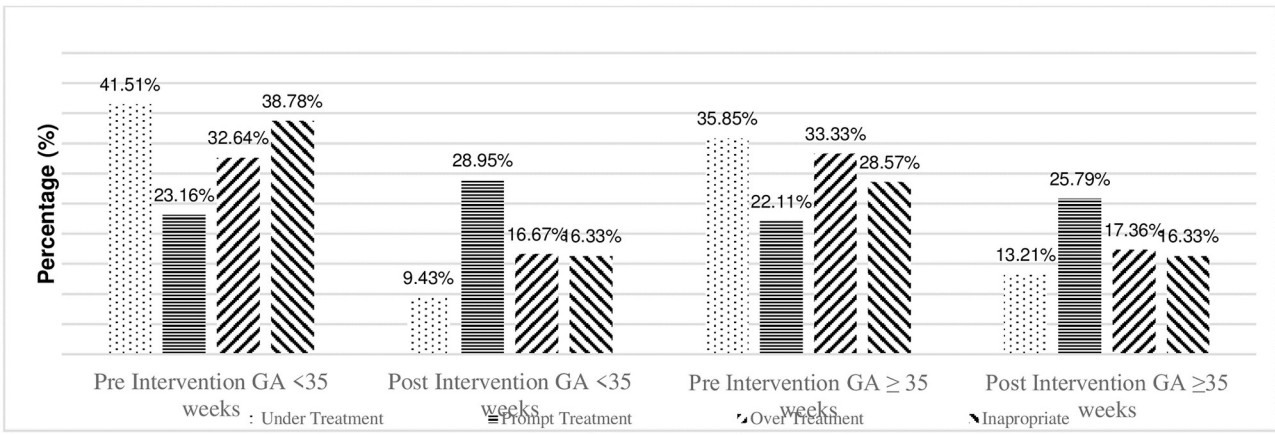

**Fig 4. Treatment classifications before and after the introduction of BiliNorm based on gestational age.** Proportions of infants with a gestational age below (grey bars) or above (black bars) 35 weeks with inappropriate treatment, under-treatment, over-treatment and correct treatment before (hatched bars) versus after (solid bars) the introduction of BiliNorm. *: p < 0.05 before versus after introduction.

introduced to the application indicated that it was perceived as helpful and easy to use. The majority of users indicated they would continue using the application. More than three-fourths of the respondents agreed that BiliNorm helped them in the management of infants with hyperbilirubinemia. This application simplified the conventional approach of

**Table 2. Patient characteristics before and after BiliNorm introduction.**

| Characteristics | All infants n = 436 | Pre-introduction n = 255 | Post-introduction n = 181 | p |
|---|---|---|---|---|
| Gestational age (weeks) | 34.7 ± 2.8 | | | 0.348 |
| <35 | | 137 (54) | 89 (49) | |
| ≥35 | | 118 (46) | 92 (51) | |
| Birth weight (g) | 2197 ± 701 | | | 0.786 |
| <1000 | | 2 (0.8) | 1 (0.6) | |
| 1000–1499 | | 22 (8.6) | 20 (11) | |
| 1500–2499 | | 149 (58.4) | 99 (54.7) | |
| ≥2500 | | 82 (32.2) | 61 (33.7) | |
| Postnatal age (days) | 4.1 ± 2.7 | | | |
| Total serum bilirubin (mg/dL) | 12.6 ± 3.9 | | | |
| Risk factors | | | | 0.614 |
| ABO/Rhesus incompatibility | | 2 (0.8) | 1 (0.6) | |
| Haemolysis: G6PD deficiency, spherocytosis, genetic predisposition. | | 0 (0) | 1 (0.6) | |
| Sick infants: asphyxia, infection, sepsis, acidosis. | | 144 (56.5) | 106 (58.6) | |
| No risk factors | | 86 (33.7) | 53 (29.2) | |
| Unknown risk factors | | 23 (9) | 20 (11) | |
| Treatment classification | | | | 0.016* |
| Correct treatment | | 97 (38) | 93 (51) | 0.006* |
| Over-treatment | | 87 (34) | 57 (32) | 0.566 |
| Under-treatment | | 35 (14) | 18 (10) | 0.234 |
| Inappropriate treatment | | 36 (14) | 13 (7) | 0.024* |

Data are presented as mean ± SD or numbers (percentages).

*: p < 0.05 before versus after the introduction of BiliNorm.

hyperbilirubinemia management and made it easy for healthcare workers. An analysis of hospital charts on the use of PT before and after the introduction of BiliNorm indicated a higher correct usage rate of PT and a less usage rate of PT without measuring the TSB level, especially in preterm infants with a gestational age <35 weeks.

Studies have demonstrated that the introduction of guidelines on how to diagnose and treat hyperbilirubinemia in newborn infants can help improve the care for newborn infants. However, the introduction of a guideline alone is not sufficient. Apparently, there is a need for campaigns to stimulate healthcare workers to use guidelines. Darling *et al* [14] explored the implementation of new guidelines formulated by the Canadian Pediatric Society in 100 hospitals in Canada and observed that 79 hospitals indicated to have implemented these guidelines. However, only 70% of these hospitals implemented measuring the TSB level before discharge, although this is recommended in the guidelines. Implementation of guidelines might help reduce the incidence of severe hyperbilirubinemia [15]. Sgro *et al.* found that after the implementation of the Canadian guidelines, the incidence of severe hyperbilirubinemia decreased from 1 in 2480 to 1 in 8352 live born infants [15].

Guidelines made for high-income countries (HIC) might not be suitable for low- and middle-income countries such as Indonesia because of the limited availability of well-equipped healthcare facilities. Therefore, a new Indonesian guideline on the diagnosis and treatment of hyperbilirubinemia was developed [7]. This new Indonesian Hyperbilirubinemia Guideline is an adoption of available international hyperbilirubinemia guidelines such as AAP guideline and NICE guideline. Previous hyperbilirubinemia guidelines from national and international institutions such as Indonesian Pediatric Society (IPS), Ministry of Health (MoH) dan World Health Organization (WHO) were evaluated and adopted into the new guideline in order to formulate a more suitable guideline to be applicated in Indonesia. The treatment thresholds to start PT are lower in this guideline than in the guidelines used in HIC. This was done to increase the safety margin, as daily control of the TSB level of a jaundiced infant is more difficult in Indonesia than in HIC. This study developed a web-based application to make the usage of this guideline easier. In a previous study, paediatricians indicated having difficulties to determine the existing guideline that hampered them to use it [3].

Comparable programs have been developed to assist healthcare workers to care for newborn infants with hyperbilirubinemia. Three of these programs, viz. BiliTool, BiliApp and the Northern California Neonatal Consortium (NCNC) app [2, 16, 17], are based on the AAP guideline of 2004 or on the NICE guideline. Most of these decision tools are applicable for infants born after ≥35 weeks of gestation, whereas others are also suitable for preterm infants. The existing decision tools only provide advices for the care of infants without risk factors. The user of these apps should make adjustments when risk factors are encountered. The BiliRecs program [18] has been designed for infants <35 weeks of gestational age, but it is unsuitable for infants with a postnatal age of <2 days.

In America, the AAP uses an application to improve compliance with guidelines made by an application known as Bilitool. However, unfortunately, Bilitool can be used only for babies with gestational age ≥35 weeks, and the reference guidelines used are from the AAP [18]. In UK, there have been efforts to improve compliance with NICE guidelines using a smartphone-based application known as BiliApp [17]. Unfortunately, although it can be used for almost all gestational age categories, the determination of gestational age in Indonesia is very difficult. This problem also exists in other low-middle-income countries. Moreover, bilirubin level is measured in the unit μmol/L, which is different from the unit used in Indonesia, i.e. mg/dL. This renders BiliApp unable to use, and hence, it would be difficult to implement it in Indonesia.

BiliNorm is different from the above-mentioned published tools. First, the treatment recommendations consider the risk factors if present. Second, the tool can be used for both term and preterm infants. Third, advices are provided regarding follow-up, based on the clinical signs of ABE in the postnatal period and on a KSD risk calculation. Fourth, BiliNorm provides information to parents about jaundice and treatment strategies. Finally, the application can be used on a smartphone. It is aimed for use in low- and middle-income countries and available in English and Bahasa languages as it is based on guidelines for Indonesia.

This study showed that after the introduction of our application in two hospitals, the incidence of correct treatment in all infants increased from 38% to 51%. Both over- and under-treatment rates of hyperbilirubinemia decreased slightly but did not reach statistical significance overall. However, >40% of the infants received treatment without good indication or did not receive treatment when indicated. The rate of use of PT without a TSB measurement decreased substantially, and it was 7% after the introduction of BiliNorm. Our study results indicate that particularly preterm infants <35 weeks of gestational age benefit the most from the introduction of BiliNorm; the rate of correct treatment increased from 33% to 53%, and that of inappropriate treatment decreased by more than half from 18% to 7%.

Limitations of BiliNorm might be caused by several factors such as; BiliNorm does not include clinical manifestation of the subject, where this factor influence clinical judgement of health care decision to do PT. It also does not involve social aspect as a consideration in hyperbilirubinemia treatment, which plays an important role especially in limited resource area. A lack of previous studies on this topic also limits theoretical foundations and comparisons related to the problem addressed in this study. More studies are required to understand why some healthcare workers either do not use or do not follow the available guidelines on the diagnosis and treatment of hyperbilirubinemia. Long-term studies are required to assess whether this application would result in improvements in the outcome of infants with hyperbilirubinemia and reductions in the incidence of severe hyperbilirubinemia and kernicterus. BiliNorm is only a decision support tool, and clinical decision is the responsibility of the clinician. As this study was conducted in two academic teaching hospitals, it might not reflect the entire implementation in daily practice where hyperbilirubinemia treatment is provided in various types of healthcare facilities that offer neonatal care.

## Conclusions

A novel web-based application known as BiliNorm was developed to improve the management of hyperbilirubinemia in Indonesia. This decision support tool is based on the Indonesian National Guideline on Hyperbilirubinemia. The results of the questionnaire survey indicated that users found the tool user-friendly and helpful in the decision to start treatment/PT and indicated that they were planning to continue using it in the future. As the use of BiliNorm improved the rate of correct treatment of hyperbilirubinemia, expanding its implementation will improve the management of hyperbilirubinemia in Indonesia.

## Disclaimer

The content of BiliNorm is designed to improve the management of jaundiced infants by qualified healthcare workers. BiliNorm is based on the Indonesian National Guideline on Hyperbilirubinemia. BiliNorm and its contents do not substitute professional medical advice, diagnosis or treatment. The use of all content of BiliNorm is entirely at your own risk. Authors of this article or the designer are not liable for any damages that may arise from the use of BiliNorm. If you are dissatisfied with this disclaimer, then the only remedy is to stop using the application.

## Supporting information

**S1 Data.**
(PDF)

## Acknowledgments

We would like to express our special gratitude to the affiliated institutions, doctors and nurses for welcoming us to join this research. We also would like to acknowledge the help of Siti Annisa Dewi Rani, MD, Abyan Irzaldy, MD, Ajeng Larasati, MD, Lutifta Hilwana, MD (Dr. Soetomo General Hospital, Surabaya), and Eko Sulistijono, MD, Paed. (Saiful Anwar General Hospital, Malang) who helped in data collection.

## Author Contributions

**Conceptualization:** Mahendra T. A. Sampurna, Pieter J. J. Sauer, Arend F. Bos.

**Data curation:** Mahendra T. A. Sampurna, Kinanti A. Ratnasari, Zahra S. Irawan, Risa Etika, Martono T. Utomo, Brigitta I. R. V. Corebima.

**Formal analysis:** Mahendra T. A. Sampurna, Kinanti A. Ratnasari, Zahra S. Irawan, Christian V. Hulzebos.

**Investigation:** Mahendra T. A. Sampurna, Kinanti A. Ratnasari, Zahra S. Irawan, Risa Etika, Martono T. Utomo, Brigitta I. R. V. Corebima.

**Methodology:** Mahendra T. A. Sampurna, Kinanti A. Ratnasari, Zahra S. Irawan, Risa Etika, Martono T. Utomo, Brigitta I. R. V. Corebima, Arend F. Bos, Christian V. Hulzebos, Peter H. Dijk.

**Project administration:** Mahendra T. A. Sampurna.

**Software:** Mahendra T. A. Sampurna, Kinanti A. Ratnasari, Zahra S. Irawan, Risa Etika, Martono T. Utomo, Brigitta I. R. V. Corebima.

**Supervision:** Pieter J. J. Sauer, Arend F. Bos, Christian V. Hulzebos, Peter H. Dijk.

**Validation:** Mahendra T. A. Sampurna.

**Visualization:** Mahendra T. A. Sampurna, Kinanti A. Ratnasari, Zahra S. Irawan, Peter H. Dijk.

**Writing – original draft:** Mahendra T. A. Sampurna.

**Writing – review & editing:** Risa Etika, Martono T. Utomo, Pieter J. J. Sauer, Christian V. Hulzebos, Peter H. Dijk.

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
