## [Decision Letter · Decision Letter 0]

18 Jan 2022

PONE-D-21-37055Evaluation of a mobile application tool (BiliNorm) to improve care for newborns with hyperbilirubinemia in IndonesiaPLOS ONE

Dear Dr. Sampurna,

Thank you for submitting your manuscript to PLOS ONE. After careful consideration, we feel that it has merit but does not fully meet PLOS ONE’s publication criteria as it currently stands. Therefore, we invite you to submit a revised version of the manuscript that addresses the points raised during the review process.

We look forward to receiving your revised manuscript.

Kind regards,

Kazumichi Fujioka

Academic Editor

PLOS ONE

Journal Requirements:

2. Thank you for submitting the above manuscript to PLOS ONE. During our internal evaluation we noted that the manuscript is also available on Research Square:

https://www.researchsquare.com/article/rs-78142/v1

PLOS supports authors who wish to share their work early through deposition of manuscripts in preprint servers and this does not impact consideration of the manuscript. However, we will not consider submissions that are currently under consideration for publication elsewhere. We note that the submission is listed as "Under review".

If your manuscript is no longer under review, we would be grateful if you could please contact BMC Medical Informatics and Decision Making/ Research Square and ask that your manuscript be updated so that it states that it is no longer under consideration by BMC Medical Informatics and Decision Making. Once this is done, please let us know so that we can allow your submission to proceed.

“This project was supported by a research grant from the National Institute of Health Research and Development (NIHRD), Ministry of Health, Republic of In-donesia, Jakarta, Indonesia HK.03.01/I/1186/2019

 which was granted to MTAS. The National Institute of Health Research and Development was responsible to provide financial assistance, as well as technical assistance through periodic review to ensure the research achieves the desired output.”

6. Please upload a copy of Figure 4, to which you refer in your text on page xx. If the figure is no longer to be included as part of the submission please remove all reference to it within the text.

Additional Editor Comments:

Fig 4 should be included in the manuscript.

Reviewers' comments:

Reviewer's Responses to Questions

**Comments to the Author**

1. Is the manuscript technically sound, and do the data support the conclusions?

Reviewer #1: Yes

Reviewer #2: Yes

2. Has the statistical analysis been performed appropriately and rigorously? 

Reviewer #1: Yes

Reviewer #2: Yes

3. Have the authors made all data underlying the findings in their manuscript fully available?

Reviewer #1: Yes

Reviewer #2: Yes

4. Is the manuscript presented in an intelligible fashion and written in standard English?

Reviewer #1: Yes

Reviewer #2: Yes

5. Review Comments to the Author

Reviewer #1: The authors developed a web-based application (BiliNorm) for management of neonatal hyperbilirubinemia. They conducted a questionnaire study and an observational study of infants with hyperbilirubinemia. The structure of manuscript is suitable. There are a few questions to the authors.

1. Page 1 line 20, “Hyperbilirubinemia” should be replaced with “Severe hyperbilirubinemia”.

2. In the Results section, the authors state 43 users of BiliNorm returned the questionnaires. However, the total number of users to whom the questionnaire was sent is not shown in Methods or Results sections. Please clarify the response rate of questionnaire.

3. The PDF of manuscript does not include Figure 4. Therefore, I couldn’t check it.

Reviewer #2: 1.General comments

BiliNorm is the first web-based decision application based on the Indonesian hyperbilirubinemia guideline. This application provided about the risks of complications due to acute bilirubin encephalopathy in addition to the advice regarding the potential treatment for hyperbilirubinemia. Moreover, BiliNorm is the advice for follow-up examination in the outpatient clinic. In the questionnaire, and most of them indicated that BiliNorm was well received and easy to use and helpful for the decision to start phototherapy. This paper appears to have been useful in the management of jaundice in newborns in Indonesia. In addition to being convenient, it seems to be very good that it can be used even in preterm infants.

2.Specific comment

a) major

・ Since the number of respondents to the questionnaire is small at 43 users, there are doubts about the effectiveness of the evaluation. It is also necessary to specify the occupation of 43 users.

・ The percentage of children who received correct treatment is shown, but the importance differs greatly between PT and exchange transfustion. It is better to show the ratio of PT and exchange transfustion among the children who received correct treatment.

・ Some users find it difficult to use BiliNorm because about 30% of the questionnaire results were bad. Therefore, the bad points of BiliNorm should be mentioned in the discussion.

・ Figure 4 is not attached, so it cannot be evaluated.

b) minor

・It is necessary to clearly state that BiliNorm is compatible with smartphones not only in the abstract but also in the method.

・It is also better to mention the penetration rate of smartphones in Indonesia.

・Since the Indonesian Hyperbilirubinemia Guideline is mentioned many times in the text, it is necessary to explain in detail.

6. PLOS authors have the option to publish the peer review history of their article (what does this mean?). If published, this will include your full peer review and any attached files.

Reviewer #1: No

Reviewer #2: No

---

## [Author Response · Author response to Decision Letter 0]

9 May 2022

Answer: Dear editor/reviewer, Thank you for your comment. We have checked our manuscript and made adjustment based on the PlosOne guidelines.

2. Thank you for submitting the above manuscript to PLOS ONE. During our internal evaluation we noted that the manuscript is also available on Research Square: https://www.researchsquare.com/article/rs-78142/v1 PLOS supports authors who wish to share their work early through deposition of manuscripts in preprint servers and this does not impact consideration of the manuscript. However, we will not consider submissions that are currently under consideration for publication elsewhere. We note that the submission is listed as "Under review". If your manuscript is no longer under review, we would be grateful if you could please contact BMC Medical Informatics and Decision Making/ Research Square and ask that your manuscript be updated so that it states that it is no longer under consideration by BMC Medical Informatics and Decision Making. Once this is done, please let us know so that we can allow your submission to proceed.

Answer: Dear Editor/reviewer, thank you for your comment. The status of our manuscript in BMC Medical Informatics and Decision is already rejected. We have also already contacted them. Therefore, it is currently no longer under consideration of any other journal. We can send you the screenshot of our submission portal that stated this manuscript is rejected

3. Thank you for stating the following financial disclosure:“This project was supported by a research grant from the National Institute of Health Research and Development (NIHRD), Ministry of Health, Republic of In-donesia, Jakarta, Indonesia HK.03.01/I/1186/2019 which was granted to MTAS. The National Institute of Health Research and Development was responsible to provide financial assistance, as well as technical assistance through periodic review to ensure the research achieves the desired output.”

Answer: Dear editor-reviewer, thank you for your comment. The funders had no role in this study, therefore we have added the sentence "The funders had no role in study design, data collection and analysis, decision to publish, or preparation of the manuscript."

Answer : Dear editor/reviewer, this manuscript contains data that potentially reflect the quality of health service of the hospitals. Therefore, data sharing is restricted unless it is really needed and require approval from multiple parties. We have stated this matter in our revised cover letter.

Answer :5. PLOS requires an ORCID iD for the corresponding author in Editorial Manager on papers submitted after December 6th, 2016. Please ensure that you have an ORCID iD and that it is validated in Editorial Manager. To do this, go to ‘Update my Information’ (in the upper left-hand corner of the main menu), and click on the Fetch/Validate link next to the ORCID field. This will take you to the ORCID site and allow you to create a new iD or authenticate a pre-existing iD in Editorial Manager. Please see the following video for instructions on linking an ORCID iD to your Editorial Manager account: https://www.youtube.com/watch?v=_xcclfuvtxQ

Answer: Dear reviewer/editor, Thank you for your inputs. We have submitted the ORCID iDs and getting it validated in the editorial manager.

6. Please upload a copy of Figure 4, to which you refer in your text on page xx. If the figure is no longer to be included as part of the submission please remove all reference to it within the text.

Answer: Dear reviewer, we have uploaded the Figure 4 and cited this figure in the manuscript accordingly.

Answer : Dear reviewer/editor, we have checked our reference list, and to the best of our knowledge it is complete and correct.

REVIEWER 1

1. Page 1 line 20, “Hyperbilirubinemia” should be replaced with “Severe hyperbilirubinemia”.

Answer: Dear reviewer, we have changed it into “Severe hyperbilirubinemia”.

2. In the Results section, the authors state 43 users of BiliNorm returned the questionnaires. However, the total number of users to whom the questionnaire was sent is not shown in Methods or Results sections. Please clarify the response rate of questionnaire.

Answer: Dear reviewer, thank you for your comment. Unfortunately, it is not possible to track the exact response rate of the questionnaire. This is because the questionnaire was given through an internal communication portal that has frequent changes in number of members. We have stated it in the manuscript in lines 142-144 and 181-183.

3. The PDF of manuscript does not include Figure 4. Therefore, I couldn’t check it. Dear reviewer, thank you for your comment. We have uploaded the Figure 4 and cited it in the manuscript accordingly.

Answer: Dear reviewer, thank you for your comment. We have uploaded the Figure 4 and cited it in the manuscript accordingly.

REVIEWER 2

1. Since the number of respondents to the questionnaire is small at 43 users, there are doubts about the effectiveness of the evaluation. It is also necessary to specify the occupation of 43 users

Answer: Dear reviewer, thank you for your comment. Unfortunately, it is not possible to track the occupation of those 43 users. The questionnaire was given in an internal communication portal which consists of pediatricians, pediatric resident physicians, and nurses.

2. The percentage of children who received correct treatment is shown, but the importance differs greatly between PT and exchange transfustion. It is better to show the ratio of PT and exchange transfustion among the children who received correct treatment

Answer: Dear reviewer, Thank you for your comment. Out of children who received correct treatment, there is only 1 child who received exchange transfusion. We wrote this information in line 215-216.

3. Some users find it difficult to use BiliNorm because about 30% of the questionnaire results were bad. Therefore, the bad points of BiliNorm should be mentioned in the discussion

Answer: Dear reviewer, Thank you for your insightful comment. We have written the limitations of the BiliNorm app in the line 291-294.

4. Figure 4 is not attached, so it cannot be evaluated

answer: Dear reviewer, thank you for your comment. We have uploaded the Figure 4 and cited it in the manuscript accordingly.

5. It is necessary to clearly state that BiliNorm is compatible with smartphones not only in the abstract but also in the method.

Answer: Dear reviewer, Thank you for your comment. In line 80 we have added that the BiliNorm can be accessed via smartphones.

6. It is also better to mention the penetration rate of smartphones in Indonesia.

Answer : Dear reviewer, thank you for your comment. We have added the information regarding the smartphone penetration in Indonesia in line 81-84

7. Since the Indonesian Hyperbilirubinemia Guideline is mentioned many times in the text, it is necessary to explain in detail.

Answer: Dear reviewer, Thank you for your comment. We have improved our explanation regarding the Indonesian Hyperbilirubinemia Guideline in line 242-255.

---

## [Editor Report · Decision Letter 1]

18 May 2022

Evaluation of a mobile application tool (BiliNorm) to improve care for newborns with hyperbilirubinemia in Indonesia

PONE-D-21-37055R1

Dear Dr. Sampurna,

We’re pleased to inform you that your manuscript has been judged scientifically suitable for publication and will be formally accepted for publication once it meets all outstanding technical requirements.

Kind regards,

Kazumichi Fujioka

Academic Editor

PLOS ONE

Additional Editor Comments (optional):

Although the uncertain response rate is main limitation, this paper is worth to be published.
---

## [Editor Report · Acceptance letter]

23 May 2022

PONE-D-21-37055R1 

Evaluation of a mobile application tool (BiliNorm) to improve care for newborns with hyperbilirubinemia in Indonesia 

Dear Dr. Sampurna:

I'm pleased to inform you that your manuscript has been deemed suitable for publication in PLOS ONE. Congratulations! Your manuscript is now with our production department. 

Kind regards, 

on behalf of

Dr. Kazumichi Fujioka 

Academic Editor

PLOS ONE